# Does supporting self-help groups of people with mental conditions for longer duration lead to more effective groups? A qualitative evaluation in Ghana

**Adam Dokurugu Yahaya[1], Lyla Adwan-Kamara[2], Peter Badimak Yaro[1], Philip Teg-Nefaah Tabong** [3]*

**1** BasicNeeds Ghana, Tamale, Ghana, **2** Ghana Somubi Dwumadie (Options Health and Inclusion Ghana LBG), Accra, East Legon, Ghana, **3** Department of Social and Behavioural Sciences, School of Public Health, College of Health Sciences, University of Ghana, Accra, Legon, Ghana

* ptabong@ug.edu.gh

## Abstract

Self Help Groups (SHGs) are informal groups amongst individuals that meet on a regular basis to discuss shared concerns. SHGs have become increasingly important in recent years for delivering development initiatives for both governmental and non-governmental organizations. To strengthen SHGs, BasicNeeds-Ghana, identified SHGs of people with mental health conditions across the country and provided them with skilled training and support. While some of these SHGs only received help for the first year, others had additional support for the second and third years. The study was conducted to compare the effects of the varying levels of support on the operations and effectiveness of SHGs in five regions in Ghana. Stratified purposive sampling technique was employed in the selection of SHGs for 32 focus group discussions (14 among SHG members who received support for year one only and 18 among those who received additional support in year two and three) and 26 interviews. The interviews were recorded and transcribed verbatim. With the aid of the NVivo 14, the data was coded using the constructs of the RE-AIM (Reach, Effectiveness, Adoption, Implementation and Maintenance (RE-AIM) framework. The results showed that SHGs that received support for more than one year were more functionally effective and performed better than those who received support for year one only. The additional training in leadership offered to SHGs in the second and third year built the capacity of members to negotiate and advocate for their engagement with governmental organizations. Regular engagements with governmental agencies motivated SHG members to attend meetings and increased their confidence to participate and engage with governmental agencies and community. The study concludes that duration of support provided to self help groups of people with mental conditions could improve the operational effectiveness. To sustain group development and to achieve growth in self-help activities, pathways for strategic support and capacity-building need to be in place at the start of the set-up.

**Data Availability Statement:** Anonymized transcripts have been uploaded as supplementary file.

**Funding:** This research was supported by funded by UKAid Grant (No: 100015) received by the second author (LAD). The funder had no role in study design, data collection and analysis, decision to publish, or preparation of the manuscript.

**Competing interests:** The authors declare that no conflict of interest exist.

## Background

Self Help Groups (SHGs) are groups of 10 to 20 women or men together who want to improve their living conditions. Membership of SHG comprises people with mutual interest or people with similar conditions e.g., mental illness [1]. The formation of SHGs has become an important component of mental health programs operated by non-governmental organizations (NGOs) in low-income countries [2]. Over the past several decades, SHGs have mushroomed as advocacy groups, alternatives, and supplements to traditional services, frequently with the goal of fostering empowerment and rehabilitation [3, 4]. A systematic review of SHGs for women in Asia, sub-Saharan Africa and the Caribbean revealed economic gains and political empowerment [5]. SHGs of people with mental health conditions are the most prevalent form of self-help. There are several thousand groups in Australia, the United Kingdom., and the United State of America (USA). In the U.S., they are utilized by over 2 million adults annually [3]. In recent time, it was reported many self-help organisations were established within the past 16 years to support the recovery of people with psychosocial disability in seven African countries.

Although the aim and dynamics of mental health self-help groups vary, they are all member-governed and place a strong emphasis on self-advocacy and actively participating in treatment decisions. These organizations are thought to offer social support and information learned from interaction with peers which addresses stigmatization and contributes towards recovery [6]. This is in line with self-attribution theories, which suggest that people with mental health conditions experience improved sense of self when they are included in meaningful groups and involved in constructive activity [7]. Self-help is also seen to reduce stigma by proving that people with mental health conditions can run their own lives and programs [8].

Governmental and non-governmental organizations are increasingly employing self-help groups (SHGs) as locations for carrying out development efforts, according to the analysis of the literature. This approach has also been used in the field of mental health to form advocacy and mutual support organizations for the members' welfare and health [1]. These SHGs in Ghana were primarily established by non-governmental organizations.

BasicNeeds-Ghana, a mental health and development advocacy organisation, implements and supports programs to improve the lives of people with mental health conditions by supporting their access to integrated social, economic, and health services in local communities around Ghana. They also provide support to build skills in advocacy and rights, and in financial training, and link groups to Metropolitan, Municipal and District Assemblies, health service institutions, social welfare and community development groups. The BasicNeeds-Ghana programme was established in northern Ghana in 2002. However, BasicNeeds-Ghana started to organize SHGs in 2006 for people with mental health conditions and those who cared for them said that being a part of groups would give them the chance to talk about issues of mutual interest and to support one another. As of 2020, 273 SHGs for people with mental health conditions have been formed by BasicNeeds Ghana across nine regions in the country. The first groups were large and consisted of people who came from widely scattered communities. Traveling to meetings of the SHGs, however, proved to be a problem. This led to restructuring to more community level groups especially in rural areas.

Before the coronavirus disease of 2019 (COVID-19) restrictions in Ghana, members of SHGs met regularly to share experiences, provide peer support to improve wellbeing, undertake livelihood activities, and engage key government stakeholders. However, following the government's directive on public assembly, their regular meetings stopped. In June 2020, Ghana Somubi Dwumadie and BasicNeed Ghana (BNGh) developed plans to support the resumption of Self-Help Group meetings, and all these 272 Self-Help Groups resumed services,

**Table 1. Provides a summary of the support that was given to SHGs.**

| Year 1 (April 2020 –March 2021)-N = 272 Groups | Year 2 (April 2021 –March 2022)-N = 28 Groups | Year 3 (April 2021 –March 2023)-N = 28-Groups |
|---|---|---|
| • Supported to reopen SHGs after COVID-19 lockdown with training on COVID-19 –causes, signs and symptoms, safety and hygiene protocols.<br>• Provided SHGs with PPEs including reusable nose masks, hand tissues, hand washing liquid soap, alcohol-based hand sanitiser, veronica buckets and it stands.<br>• Trained SHGs on records keeping and support them with stationery (pens, attendance register, rulers, endorsing ink, stamp pad, stapler, ruled foolscap papers, arch file, staple pins, etc.)<br>• Trained SHGs on COVID-19 vaccine awareness<br>• Trained SHGs on safeguarding<br>• Supported SHGs to engage their MMDAs to demand inclusion / access to social protection interventions such as DACF, NHIS, LEAP, etc for members of SHGs, especially women | • Organized refresher training on Safeguarding and reporting to community volunteers.<br>• Facilitated follow up interface meetings between representatives of SHGs with key staff of MMDAs for improved support to persons with mental health conditions.<br>• Trained SHGs on group development and leadership<br>• Supported community mental health volunteers to undertake home visits and provide MHPSS services to SHGs in their communities and provide monthly reports to BNGh | • Trained SHGs on conflict management, challenges women in leadership roles face<br>• Refresher training on safeguarding.<br>• Facilitated follow up interface meetings between representatives of SHGs with key staff of MMDAs for improved support to persons with mental health conditions. |

reaching 4,366 members in Northern, Northeast, Savannah, Upper East, Upper West, Bono, Bono East, Ahafo and Greater Accra Regions and provided them with support for one year. After the first year, some of the groups were selected to receive additional capacity building support for two more years (Table 1). To select which SHGs received additional support in year 2 and 3, six regions were purposively selected from a total of nine, representing both Northern (Rural) and Southern (Urban) areas where programme interventions were implemented. Within each of the six regions, all districts with Self-Help Groups (SHGs) were identified, and districts were randomly selected. Additionally, SHGs within the selected district were listed on folded pieces of paper, and one SHG was randomly selected. Regions with a greater number of districts had a larger number of districts and, therefore, more SHGs included in the selection. The effectiveness and functionality of SHGs are now affected by a variety of support systems. This study was therefore conducted to evaluate the effects of varying level of support on the operations and effectiveness of SHGs in Ghana. This is relevant to fill the gap in literature and provides evidence to improve the formation and operations of SHGs for people with mental health conditions.

## Methods and material

### Study design

This study used a narrative qualitative research design [9] within the framework of implementation research (IR). This study adapted the Reach, Effectiveness, Adoption, Implementation and Maintenance (RE-AIM) framework of implementation research. The RE-AIM is a planning and evaluation model that addresses five dimensions of setting level outcomes important to Program impact and sustainability—Reach, Effectiveness, Adoption, Implementation, and Maintenance [10]. Reach refers to the absolute number, proportion, and representativeness of individuals who participate in each intervention or Programme, in this case SHG activities. Effectiveness is the impact of an intervention on important outcomes and includes negative effects, quality of life, and economic outcomes. Using the RE-AIM framework, effectiveness of the intervention was measured to determine the contribution of the SHGs to the welfare of people with mental health conditions and disability. It also examined the engagement with other governmental institutions. Adoption is the absolute number, proportion, and representativeness of settings and intervention agents who initiate a program. This was explored by

examining the activities of the SHGs and collaborations with government agencies and departments in decision making. The study also measured the context (Adoption) in which their activities were conducted, highlighting the facilitators and barriers as well as sustainability strategies. This was relevant in knowledge translation. Implementation refers to the intervention agents' fidelity to, and adaptations of, an intervention and associated implementation strategies, including consistency of implementation as intended and the time and costs involved. Lastly, maintenance is the extent to which a Program or policy becomes institutionalized or part of the routine organizational practices and policies. In this study, the sustainability of the SHGs was explored. Within the RE-AIM framework, maintenance also applies at the individual level, and has been defined as the long-term effects of a program on outcomes after six or more months of intervention contact. Since the SHGs are receiving support from Basic-Needs-Ghana, which has the database of members, the "Reach" construct of the RE-AIM model was not utilized.

## Study area

The study was conducted in five regions in Ghana, namely Northern, Northeast, Savannah, Upper East and Greater Accra (Fig 1). These regions have SHGs that have received support from BasicNeeds Ghana (BNGh), as part of Ghana Somubi Dwumadie consortium, led by Options Consultancy Services and including BNGh among others.

## Study population

The population for the study included members of SHGs located in the five selected regions that received support from BasicNeeds Ghana, stakeholders involved in the provision of services for people with mental health conditions and disabilities in the selected regions.

## Selection of participants

Stratified purposive sampling technique was employed in the selection of study participants. In each region, SHGs were stratified into two; those that received support for year one and those who continue to receive additional support in year two and three. Proportional numbers were assigned to each group. Following that purposive sampling was used to select members who had good knowledge on the operations of the groups.

## Data collection strategies

Two main data collection strategies were employed in this study; focus group discussions and in-depth interviews.

## Focus group discussions

Focus group discussions was organized among members of SHGs. SHGs were put into two groups (Year one only, >Year 1). In all, more mixed FGDs were organized. This was done to ensure that the minimum requirement of six members in FGDs was achieved. For FGDs with people having the same condition, mixed gender groups are permissible [11]. At least one FGDs was conducted in each of the five regions per category; received support for only one year and more than one year. Each FGD comprised of at least seven participants (Table 2).

## In-depth interviews

In-depth interviews were conducted with leaders of the SHGs—chairperson, vice chairperson, secretary, treasurer, and organiser. In addition, eight selected stakeholders such as

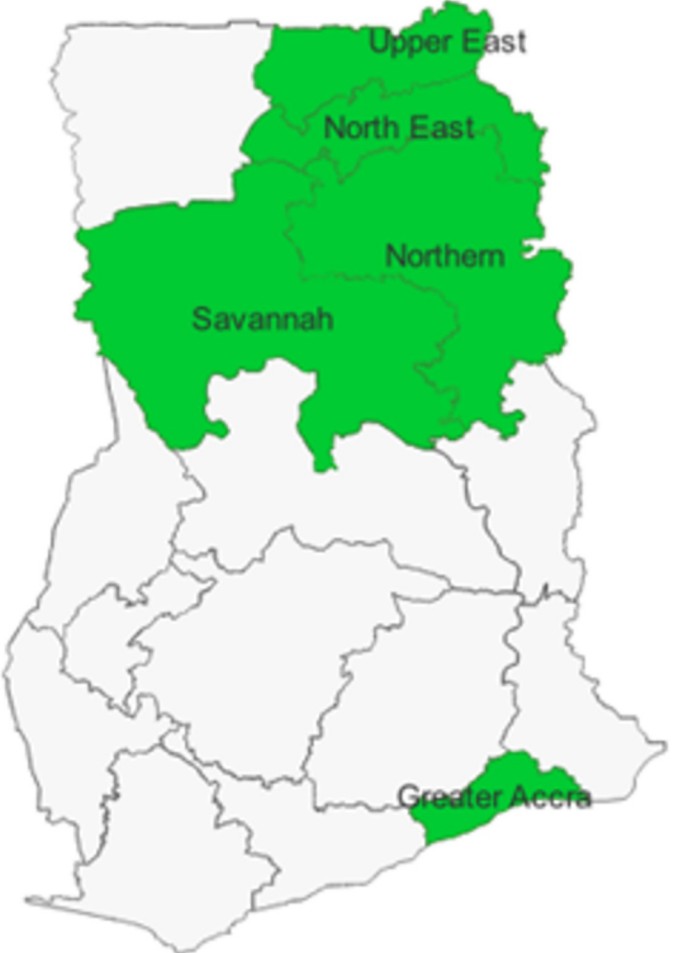

**Fig 1. Map of Ghana showing study regions.** Source: Created with ArcGIS (basemap shapefile: Ghana Statistical Service: https://data.humdata.org/dataset/cod-ab-gha).

representative of Metropolitan, Municipal, District Assemblies, (MMDAs) Department of Social Welfare, or National Health Insurance Scheme. Maximum variation purposive sampling techniques was employed to ensure diversity in the stakeholders that participated in the study. In all, 20 in-depth interviews were conducted with leaders of SHGs in the study. Two groups were selected in each region per category (year 1 only, > year 1). In each one of the executives were interviewed (Table 3).

**Table 2. Distribution of FGDs.**

|  | Year 1 only | >Year 1 |
| --- | --- | --- |
| Males | 2 | 1 |
| Females | 3 | 2 |
| Mixed Group | 2 | 6 |
| Total Number of FGDs | 7 | 9 |
| Total Number of Participants | 61 | 72 |

**Table 3. Qualitative sampling.**

| Executive Category | Year one only (number of interviews conducted) | >Year one year (number of interviews conducted) |
|---|---|---|
| Chairperson | 2 | 2 |
| Vice chairperson | 2 | 2 |
| Secretary | 2 | 2 |
| Treasurer | 2 | 2 |
| Organizers | 2 | 2 |
| Total | 10 | 10 |

In addition, stakeholders such as representative of Metropolitan, Municipal, and District Assemble (MMDAs)-3, Department of Social Welfare / Community Development-2, National Health Insurance Scheme-1 participated in the interview segment.

## Data collection tool

Interview and focus group discussion guides were developed for data collection. Conceptual dimensions of the RE-AIM model were used as the thematic areas for the in-depth interviews. These included social protection interventions for people with mental health conditions and disability, operations of the SHGs, funding sources, collaborations and engagement with governmental organizations, and sustainability of activities of SHGs. The FGDs guide also covered similar thematic areas.

The interview guides were developed in English and translated into the local language (Dagbani, Mamprusi, Frafra, Gonja and Ga) during interview sessions. Training was organized where the data collectors translated the English versions of the various topic guides and agreed on the most appropriate way of asking the questions. This was done to ensure consistency in the data collection.

## Data collection procedure

Both FGDs and IDIs were conducted on a face-to-face basis. Interviews for SHG members were conducted by trained graduate research assistants whilst that of the stakeholders was conducted by the corresponding author. The FGDs were conducted by a trained moderator with support of a notetaker. The moderators had expertise in the conduct of FGDs and were linguistically competent. FGDs and interviews with leaders of SHGs were conducted in the community at the venue where their meeting are often held. However, interviews with stakeholders occurred at the offices of the selected individuals. All interviews were recorded using a digital voice recorder. While FGDs lasted 60 minutes, each interview took between 30 and 45 minutes. The data collection took place between August and September 2023. During FGDs, questions were posed by the moderator and all participants were made to actively discussed each thematic areas to arrive a consensus or a group position. This was done to ensure that what was finally arrived at during the discussion fairly represented the reality of the group with respect to the thematic areas of the RE-AIM model. After each discussion or interview, the moderator summarised the key highlights as a form of member checking [12].

## Data analysis

The interviews were transcribed verbatim. Interviews and FGDs conducted in local languages were translated during transcription. NVivo 14 was used for the qualitative data analysis. Thematic analysis was employed in this study [13]. Hybrid inductive-deductive (Abductive)

approach was used to develop the codebook. Developing a codebook using this approach involves two stages. First, prior to data collection, a codebook was developed using conceptual dimensions of the interview guides. As such, the study adopted the constructs in the RE-AIM model to develop the codebook. This was revised after data collection to capture new information (second stage). Double coding was done, and code comparison query ran in NVivo. The intercoder reliability index (Kappa Coefficient) was computed as 0.88 which indicates higher level agreement [14]. Coded sections were regrouped into relevant categories and themes for presenting the results. Direct quotations were used, where appropriate, to support the themes. Finally, data triangulation was done to merge the findings from the various data sources.

## Ethical approval

The protocol of the study was reviewed and approved by the Ghana Health Service Ethics Review Committee for approval. All requirements for conducting studies using human subjects were complied with in this research. All the participants either signed or thumb printed an informed consent form before the study.

## Results

### Main and subthemes

Based on the data analysis and the constructs of the RE-AIM model which was adoped for this study, the main and subthemes have been summarised on Table 4.

### Effective performance and functionality of SHGs

This included participation in meetings, engaging with government agencies and community activities and contribution of the group to the welfare of members. In both FGDs and IDIs with members of SHGs that received support for than one year and IDIs with their leaders, it emerged clearly that all of them organize monthly meetings and had provided minutes (evidence) to support discussions. One group leader stated during an IDI in the Greater Accra region as follows:

> "*We used to meet every Thursday but it wasn't favouring the majority at a point and so we changed the meeting days to last Thursday of every month and the members participate a lot during meetings*" **(Male, IDI, Leader, GAR)**

**Table 4. Main and subthemes.**

| Main themes (Constructs of RE-AIM Model) | Sub-themes |
|---|---|
| Effectiveness (contributions of SHGs to welfare of members) | Organizing meetings |
| | Participation in meetings |
| | Engagement with government agencies |
| | Advocacy |
| Adoption | Engagement with members |
| | Engagement of government agencies |
| | Demand for human rights by SHGs |
| Implementation | Organising health education and campaigns |
| Maintenance/Sustainability of activities of SHGs | Continuous training |
| | Providing support |
| | Collaboration with community volunteers |

However, in interviews with leaders of SHGs that received support for just one year, they lamented the attrition of members and poor attendance of meetings. One leader revealed in an interview:

"*Our membership keeps reducing and people do not come for meetings regularly. So sometimes we the leaders even do not want to organize the meetings*" **(Female, IDI, Leader, North East)**

Furthermore, the study found that SHGs that were supported for more than one year held regular engagement and reported they were able to build stronger relationships among members as well as with government agencies. In addition, members of SHGs that received support for more than one year indicated motivation was high among members in attending meetings and engagement with government agencies. Two participants of FGDs provided an explanation as follows:

"*BasicNeeds [Ghana] was able to support us to engage MMDAs and the Social Welfare Department. So, we now have a well-established and stronger relationship with these institutions. Our members are also happy anytime we have such meeting. So, it is making the group stronger*" **(Male, FGD, Savannah, >Year 1)**

"*. . .we have a stronger relation among group members and the government institutions. So, when you hear of upcoming meeting that you think has something to do with people with mental illness [mental health conditions], we call them to include us in invited guest list. This is making our group stronger. Members feel they are important, that is why we are invited to participate in such meetings. The motivation to attend meetings or engagement with government agencies is high among our group members*" **(Female, FGD, GAR, >Year 1)**

Moreover, SHGs members indicated that the leadership training they received in the second and third year has improved their advocacy skills and techniques. This has therefore led to better engagement with governmental institutions such as the MMDAs, DSW/CD and health institutions. Participants also revealed that the training has built their confidence to be able to engage effectively. The following illustrative quotes buttress these points:

"*We had training on leadership and advocacy. This has therefore empowered us to able to negotiate very well. When you attend meetings and show good leadership qualities in your contribution, next time they will invite you*" **(Female, Leader, Upper East region, >Year 1).**

"*Now when we participate in meetings, we have the confidence to advocate for the rights of our members. This is very important benefit we got from the support we received from BasicNeeds [Ghana]*" **(Male, Leader, Northeast region)**

### Adoption of Self Help Groups (SHGs) activities

With regards to the adoption of the activities, both SHGs who received support for just one year and more than one year reported some level of engagement of members. However, the data suggests that SHGs that were supported for more than one year indicated the regular engagement was able to build stronger relationships among members as well as with government agencies. In addition, members of SHGs that received support for more than one year indicated motivation was high among members in attending meetings and engagement with government agencies. One participant of FGDs provided an explanation as follows:

"*. . .we have a stronger relation among group members and the government institutions. So, when you hear of upcoming meeting that you think has something to do with people with mental illness [mental health condition], we call them to include us in invited guest list. This is making our group stronger. Members feel they are important, that is why we are invited to participate in such meetings. The motivation to attend meetings or engagement with government agencies is high among our group members*" (**Female, FGD, GAR, >Year 1**)

Leaders in groups that received support for more than one year corroborated the findings of engagement in advocacy for registration in the National Health Insurance Scheme. This is supported by the following quote:

"*We met with the Department Social Welfare I think early this year and NHIS also met with us some time ago I can't remember they registered our members into the NHIS and also renewed some of us our cards*" (**Nima, Female, > Year 1, GAR**)

In interviews with some leaders, it was revealed that involving members of SHGs in decision making especially those that directly affect the health, and social wellbeing was a right that had to be respected by all institutions in the country. They emphasize that engaging people with mental health conditions and disability was a right enshrined in international and national conventions and laws. For example, one leader in SHG that received support for more than one year illustrated:

"*In Ghana, people think engaging people with mental illness [mental health conditions] and disability in decision making is a privilege, but it is our right. This has clearly stated in Ghana's Disability Act*" (**male, IDI, leader, GAR**)

Another leader in SHG that received support for than one year referred to the United Nations Convention on the Rights of Persons with Disabilities to assert their rights to participate and be engaged in decision and policy making. She however lamented successive governments inability to protect those rights as follows:

"*Ghana is signatory to the United Nations Convention on the Rights of Persons with Disability, but the country is not able to implement the requirements. Institutions and government agencies sometimes do not engage us in decision making. It is disappointing*" (**male. IDI. Leader, GAR**)

In interviews with a district coordinating director, he admitted that there was the need to regularly engage people with mental health conditions and other disabilities in decision making to ensure fair representation of vulnerable groups.

"*People with mental illness and disability are a critical mass in Ghana and should be engaged in decision making and policy formulation. Our constitution and the Disability Act require fair representation in decision making*" (**Municipal Coordinating Director, IDI, GAR**).

### Implementation: Perceived effects of engagement on psychosocial wellbeing of members

The study found that engaging people with mental illness in the community activities has positive psychosocial effects and reduces stigmatization of the people with such disabilities. In

FGDs, SHG members were unanimous in expressing their delight on how engaging them in community activities has contributed to creating a sense of belonging as well as increasing the ability to interact with others and contribute during meetings. In the IDIs with leaders, it was emphasized that engaging members of SHGs in community activities was instrumental in community acceptance of people with mental illness and disability, hence reducing stigma associated with mental health conditions and disabilities. This view was expressed by a leader of a group that received support for more than one year as follows:

"*We are happy when we involved in community activities... when people involve us in their activities, and it has helped us to now socialize with people in the community more unlike before when we were introverts, and we can't come around or associate with people. It also reduces stigmatization of our group members*" **(Group Leader, Male, One year, GAR)**

In addition, a SHG members revealed that they organize health campaigns to educate community members on the causes of mental health conditions. This, in the opinion some participants, had addressed the widely held misconception that mental health conditions were a spiritual condition:

"*Some of the benefits of health campaign we organized has create the awareness to people who think their condition is spiritual to now believe that mental health condition can be manage through medical treatment and now they have become well*" **(IDI, female, >year 1, GAR)**

However, for SHGs that were currently not receiving support, it was clear leaders expressed concern about member absenteeism during meetings. Some leaders indicated they had to move from house to house to remind people about the meeting. This is because when they come, they expect something, and some incur cost in moving to venues for the meetings. This was expressed as follows:

"*Sometimes it is difficult to converge them at the meeting place. I have to move from house to house to remind them about our meetings, but some are not interested. The enthusiasm was there when we were receiving support but now it is a challenge*" **(Group Leader, GAR)**

The lack of additional support in the view of SHGs that received support for year one only led to the attrition of members of the SHGs. This was especially so among SHGs who were currently not receiving support as illustrated:

"*Our membership rose to 70 plus and as at now we have reduced to 40 plus as some people have stopped coming for meeting because of the lack of financial and training support*" **(Female, FGD, Year 1 only, Savannah Region)**

The results also showed that stigmatization is a barrier to participation in community activities. This view was largely expressed by members of SHGs that have received support for year one only. One participant shared her view as follows:

"Some of our members do not want to go out because of the stigma against people with mental health conditions in this community. This is one of the reasons some people refuse to attend meeting. So, you call for the meeting and just a few people come. So, it is discouraging" **(Female, FGD, Year 1 only, Northeast).**

In contrast to this, a member of SHGs that received support for than one year revealed they organized community sensitization activities which contributed to the reducing stigmatization of people with mental illness. In a FGD with SHG members who have received support for more than one year, this was a well-entrenched position among participants. One participant shared her views as follow:

*"We organize community durbars in the community and educate people about mental health conditions. At first, people had the belief that mental health conditions was spiritual and when you come close or in contact with people with the condition, you can become infected"* **(male, FGD, Upper East)**

## Implementation of activities of SHGs

In interviews with some leaders, it was revealed that involving members of SHGs in decision making especially those that directly affect the health and social wellbeing was a right that had to be respected by all institutions in the country. They emphasize that engaging people with mental health conditions and disability was a right enshrined in international and national conventions and laws. For example, one leader in SHG that received support for than one year illustrated:

*"In Ghana, people think engaging people with mental illness and disability in decision making is a privilege, but it is our right. This has clearly stated in Ghana's Disability Act"* **(male, IDI, leader, Year 1 only, Accra)**

Another leader in SHG that received support for more than one year referred to the United Nations Convention on the Rights of Persons with Disabilities to assert their rights to participate and be engaged in decision and policy making. She however lamented successive governments inability to protect those rights as follows:

*"Ghana is signatory to the United Nations Convention on the Rights of Persons with Disability, but the country is not able to implement the requirements. Institutions and government agencies sometimes do not engage us in decision making. It is disappointing"* **(male. IDI. Leader, > Year 1, Accra)**

In interviews with a district coordinating director, he admitted that there was the need to regularly engage people with mental health conditions and other disabilities in decision making to ensure fair representation of vulnerable groups. This position has been supported by the illustrative quote:

*"People with mental illness and disability are a critical mass in Ghana and should be engaged in decision making and policy formulation. Our constitution and the Disability Act require fair representation in decision making"* **(Municipal Coordinating Director, IDI, Accra)**.

## Sustainability of SHGs activities

Participants indicated the support they received is largely from non-governmental organization for both SHGs that have received support for year one only and >year one. Some leaders shared her experience as follows:

"*We don't receive support frequently, support come when we meet with NGOs (Non-Governmental Organization). Sometimes we get funds from a benevolent lady who pays some of our members' hospital bills when they go to health facilities and their medications*" **(Male, Leader, IDI, Year 1 only, GAR)**

"*The last time we received support was early this year from an individual benevolent person who gave two of our members funds to start trade and BasicNeeds Ghana also gave us funds and an individual woman called Ann gave us money during the Ramadan festival*" **(Female, Leader, IDI, >Year 1, Northeast Region)**

One way of sustaining the operations of SHGs was contributions from the members. A member in a focus group discussion in North East region shared her experience as follows:

"*Where we usually get support from is, sometimes, within this group. We usually contribute money any time we meet and it has helped me, I have people here who can testify that when BasicNeeds-Ghana came and gave us preventive stuff [Nose mask, hand sanitizer] and it got finished we used the money we had in our account to continue buying the sanitizers and the hand wash, so anytime we have met before we sit for the meeting, we wash our hands first*" **(Male, FGD, Year 1 only, North East)**.

In the Greater Accra region, in IDI with a leader of a group that received support for than one year, she stated as follows:

"*Initially we used to contribute little money like 50 pesewas and 1 cedi and give it to one person to start business with and later that person pays it back to us and we will give the payment to another person to also trade with it.*"

Supporting members to engage in trading was suggested during focus group discussions. One participant shared her experience selling cooking ingredients as follows:

*After my husband rejected me and my children, I didn't relent in my effort by looking somewhere for support but invested the little I have into petty trading from house to house every morning till evening. I have now extended the selling to the suburbs of the community. Today I acquired a building plot and am able to put 4 bedrooms that I'm occupying with my children. The children are now grown and working. I made a garden in my house and that one alone gives us some income*" **(Female, FGD, Savannah Region)**

In FGDs with SHGs that received support for only year one, it emerged that they had to encourage their members to contribute to a mutual fund monthly to support their activities because they were no longer receiving regular support.

"*Our group members met and decided to start a monthly contribution to support ourselves. Support from NGOs and individuals is not regular. So instead of waiting for support, we decided to do something for ourselves*" **(Female, FGD, Year 1 only, Greater Accra)**

Some SHGs also indicated that they work closely with community health volunteers. In communities where volunteers were described as active, SHGs generally had active members and regularly organized meetings and participated in community activities. This was unanimous across the IDI and FGD participants for groups that received support for only year one

and those who received additional support for year two and three. The following illustrative quotes support this assertion:

> "*Our community health volunteers are very supportive, I can say that we have active members because of the volunteers regularly engage us in community activities*" (Male, FGD, Year 1 only, Savannah)

One leader highlighted the challenges in attending meetings especially in urban areas as follows:

> "*I can say that our meeting once in every month but the participation is not all that good because in my group like this not everyone in the Tebibianor district comes to the meetings those who are a bit far away like Spintex find it difficult to come because of money issues, it is not a walking distance whereby they can walk to meetings and so they are asking for financial support to help them come to meeting regularly*" **(IDI, > Year 1, Accra)**

To address the financial challenges for people with mental health conditions, it was recommended that the government should consider the inclusion of People with Mental health conditions in the District Assembly Common Fund. Although the inclusion has been done, none of the SHGs member seemed to be aware. In a FGD with a SHG members, one participant said the following:

> "*When we started and went to District Assembly and asked for our 3% of the DACF they told us about that people with mental health conditions and disability were not part of the beneficiaries of the fund. It's rather the physically challenged, not mental health patients and that did not make us happy*" **(Male, FGD, Northeast)**

### Inclusion of medicine for mental health conditions into national health insurance scheme benefits

Across all the groups (year 1 only and > year one), it emerged that concerns were raised on the urgent need to include medicines for people with mental health conditions into the scheme. According to SHG members, this would help address the shortage of such medicines in hospitals. The following quotes support this call:

> "*The challenge we encountered was that when we spoke with the nurses about the drugs, they said government has not supplied them with drugs and if you are someone with frequent seizures and goes to hospital and you are told there is no drugs, do not think that the drugs are there but they refuse to give you. So, you the patient would have to find out where drugs could be acquired so you go and buy so the nurses can treat your patient for you*" **(Female, FGD, > Year 1, Northeast)**

> "*The government must initiate policy that will give people with mental health issue the freedom to work, and the government should also ensure that our medicines are also included in the health insurance*" **(Female, leader, Year 1 only, GAR)**

Participants lamented the negative effects of the shortage of medicine on their health as well as financial implications. According to participants, they were sometimes compelled to do out-of-pocket for the medicines because they had to get them from pharmacies. The following buttress these points:

"*When we go to hospital, we are always told that the medicines are not available so you should go and buy from a pharmacy. Given that majority of us are not working, we are unable to afford. So, some people may stop taking the medicine because they cannot buy it and their health would deteriorate*" (**Female, IDI, Year 1 only, GAR**)

"*I have health insurance card but anytime I go for review, I must buy the medicine, but we don't have money. Why do you register us for the national health insurance for free yet, when we are sick or go for review, we have to pay. It is becoming difficult for us*" (**Male, FGD, > Year 1, northeast**)

The findings show that the lack of psychotropic medications in health facilities was negatively affecting the performance and functioning of SHGs. Participants were of the view that members who relapse because they could not afford to pay out-of-pocket for their medications automatically leave the group when their condition relapse. This has led to attrition of members of the group. In addition, members were concerned that their needs have been neglected after several calls for the inclusion of psychotropic medications on the health insurance scheme. Two group leaders shared their view on this as follows:

"*Some of our members get frustrated when they go to hospital and are not given the medications. This demotivates them and make some unwilling to attend meetings because they think that government do not care about their concerns. So, it was waste of time to attend meeting and engage government institutions*" (**Male, IDI, Leader, > Year 1, GAR**)

"*In fact, the lack of drugs for us [people with mental illness] is negatively affecting the performance of our members. It is demoralising that several years after we have been advocating for the government to add medications for mental [health] conditions into the health insurance, it has still not been done. Some of our members have refused coming for meetings because their condition has become worse*" (**Female, IDI, Leader, Year 1, Northern**)

A manager of a national health insurance scheme acknowledges the urgent need to include psychotropic medication into the benefit. He assured that plans were afoot to include medicines for people with mental health disorders in the scheme.

"*It is time the psychotropic medications are added to the scheme, both the President and Executive Director of NHIS are committed to the cause and have made public statements to that effect. Very soon those drugs would be included*" (**District Manager, NHIS, GAR**)

In summary the study identified several challenges among both SHGs that received support for year one only and those who received support for more than one year. These challenges and the suggested solutions have been summarised in Table 5.

## Discussion

### Effectiveness of SHGs in engaging with governmental agencies and influence of policy

SHG engagement with governmental agencies is not only critical in promoting the health and social wellbeing of members but provides an opportunity to influence policy positively. Both self-groups that have been provided support for just one year and those who received support beyond that indicated some level engagement. It was clear from this study that these engagements are occasionally for SHGs that received support for year one only. The supplementary

**Table 5. Challenges and suggested solutions.**

| Challenges | Suggested solutions | |
|---|---|---|
| | **Programme Level** | **Policy Level** |
| Financial | • Financial support to start trading.<br>• Skilled trading and provision of start-up capital | • Inclusion of people with mental illness in District Assembly Common fund (DACF) |
| Access to medicines | • Registration on NHIS | • Inclusion of medicines on NHIS |
| Transportation to meeting ground | • Support to attend meeting.<br>• Reducing the frequency of meetings<br>• Creating smaller sub-groups in communities | |
| Stigma and discrimination | • Community sensitization on mental illness<br>• Health education | |

training on leadership and negotiation skills that was provided to SHGs in year two and three made the members more confident and better positioned to negotiate with government institutions. This underscores the need for continued engaging of SHGs in health and developmental activities that affect their lives.

Engaging people with mental health condition and disability is a right enshrined in Ghana's Disability Act of 2006 [15]. Ghana can learn from other countries that engage people with disability in all policies affecting them. In South Africa, in April 2012, the National Ministry of Health hosted provincial consultations and a national summit on mental health policy and legislation. Stakeholder perspectives in the draft mental health policy presented at the summit included the views of service users [16].

The growth of peer support and advocacy in Lower-and Middle-Income Countries (LMICs) raises the possibility of fostering solidarity that could lead to the development of group actions to assert their rights. These have, however, only had a minor impact on mental health policy thus far, and efforts are impeded by a lack of funding. Additionally, there may be room for cultivating community solidarity, building on the social commitments shown by individuals who helped others who were struggling with mental illness find work. Church organizations, youth clubs, labor unions, and professional societies are examples of community networks and resources that are grounded in natural social values and connections, in contrast to NGO programs that are frequently transient, unevenly dispersed, and driven by outside financing and objectives. A related study showed that SHG members who participated in local governments activities indicated that it increased their confidence and facilitated their taking up leadership positions [17].

Nonetheless, the present study revealed SHGs members expressed concerns about the tendency of community members to stigmatize and discriminate against people with mental health conditions. This stigma was reported to be high among patients with mental health conditions because of misconceptions about causes. These misconceptions created a situation where community members overestimated the risk of transmission and negative attitude towards people with the condition. Over-estimation of risk of transmissibility of conditions have often resulted in discriminatory attitude towards people with such conditions [18, 19]. Nonetheless, SHGs that received support for more than a year managed to organize community sensitization and education sessions as platforms to interact and engage with community members. The community engagement was found to be effective in reducing stigma and discrimination as reported in an earlier study [20].

The study underscores the need for interventions aimed at improving knowledge about mental health conditions and targeting the misconceptions that have been identified. Health education on mental health conditions using cultural, and community appropriate strategies are required to provide the community with the right information about mental health conditions to increase knowledge and disabuse the misconceptions. Acquiring correct information about mental health conditions has the potential to improve health seeking behavior. An earlier study in Ghana reported that stigmatization of deaf served as barrier to seeking professional psychiatric care [21].

## Adoption of SHGs activities

The study findings clearly showed that SHGs supported for more than one year were able to organize regular meetings with good active membership. A stable membership, committed to the group's activities, was evident in the regular attendance of weekly meetings and the commitment of team members in SHGs that received support for more than one year. Stable and active membership in critical in the survival of the SHGs. In a study in Kilifi, Kenya, was reported that SHGs which were not receiving regular support became inactive. The study reports that seven groups out of the 18 registered groups had disbanded after the seven months without support [22]. The stable memberships in SHGs who received additional support in year two and three were largely due to the additional capacity building training in negotiation and leadership. It is therefore important to build the capacity of members of SHGs given the important role they play in the socio-economic and psychological wellbeing of its members. Earlier study supports the contention that self-help initiatives can serve as valuable vehicles for participation of users in their own and others' recovery processes [23]. Self-help groups often create genuine community support systems which enhance and supplement existing health and mental health care systems.

Nonetheless, the cost of transport to attend meetings regularly emerged as a barrier, especially in urban areas. This has therefore led to some groups becoming inactive. Transport limitations and a lack of finances would also likely have affected their attendance [24]. Paying regular monitoring visits to SHGs by government agencies and health workers can help encourage members to regularly attend meetings. As it has been reported, compliance with monitoring visits appeared to be critical to group survival and growth of SHGs [24]. To improve the financial status of members, some SHGs make monthly contributions. This study findings underscores the need for SHGs to be encouraged to engage in monthly contributions as this is a more sustainable approach. In a related study, it has been reported that developing the skills of SHG members presents the best opportunity to empower the group and ensure sustainability [25]. Relying on support from Non-Governmental and Civil Society Organisations is good but not sustainable. Government must find a way to support the activities of SHGs given the critical role they play in socio-economic welfare and health of members.

## Sustainability of activities of SHGs

For the SHGs to be able to sustain their activities, it was suggested measures should be Put in place to ensure improvement in the condition of people with mental illness. One such measures was the availability and regular supplies of medications. As Given the fact that Frequent shortages were reported. These shortages were attributed to the non-inclusion of Psychotropic medicines in the national health insurance scheme. Due to shortages of psychotropic medication in public hospitals, people with mental health conditions and their relatives are often compelled to purchase them out of pocket. But since they worked in low-wage jobs, they were unable to afford such treatment in the long term, and the precarity and struggle to survive

negatively affected their mental health. For patients who cannot afford to pay out-of-pocket for their medicine, they are compelled to stay without medicine which can lead to relapse, which can negatively affect their participation in group activities and can affect the performance of SHGs. This underscores the need to ensure that medicines are provided to biomedical facilities to improve upon mental health care. The inclusion of the essential medicines for mental health conditions into the NHIS could help address the periodic shortage and take away the financial catastrophic effect on patients with mental health conditions and their caretakers. The absence of medicine at biomedical facilities also favours and pushes people towards traditional and spiritual health care. However, traditional, and spiritual healers are often ill-equipped to effectively manage mental illness, hence the need for collaboration with health care facilities [26].

In March 2012, the Parliament of Ghana passed the Mental Health Act, 2012 (846). Some of the key areas of the law include the establishment of a mental health authority to be responsible for policy direction, service quality standards, community-oriented care, enshrinement of human rights, and least restrictive environment for people with mental illness. It also included strong emphasis on public education, integration of mental health into the general health care, establishment of mental health fund, free mental health care to all in need, wide accessibility to quality mental health services. However, the exclusion of the essential medicines on NHIS list may undermine access to care for people with mental health condition as expressed by participants in this study. An analysis of the Essential medicines and NHIS Lists by Ghana Somubi Dwumadie also raised similar concern on the exclusion of medicines for mental disorders [27]. In addition, an earlier study has reported that the shortage of medicines has led to the use expired drugs [26]. Given the participants in this study were registered with NHIS, the inclusion of medicines would help reduce the financial catastrophe of out-of-pocket payments and improve their functionality and performance.

The study findings also underscore the need for sustainable funding of the activities of SHGs. To achieve this, it was suggested that mental health condition should be included in the District Assemble Common Fund (DACF). Aside from legislation, Ghana has two social protection programs that, in principle, should be available to people with mental illness who are unable to work. The Livelihood Empowerment Against Poverty (LEAP), a cash transfer program adopted in 2008, provides small monthly payments to people with "severe" disabilities. Secondly, a proportion of the "Common Fund" is allocated to local government to be made available to people with disabilities. In practice, these funds are difficult to access and inconsistently distributed [28]. An earlier study reported that most persons with disability at the district level did not know the specific provisions of the DACF. As such MMDAs continue to breach and abuse of administrative powers in the distribution [29]. Some mental health workers and NGOs such as BasicNeeds-Ghana work with Social Welfare officers to help people with mental health problems apply to their local district assembly for these funds, but most of those interviewed did not receive them.

The inclusion of people with mental health conditions in the 3% disability fund portion of the district assembly common fund could potentially improve the livelihood of such people and could also be used to provide continuous capacity strengthening activities. Again, economic empowerment of these people would not only improve their quality of life but reduce mental disorders as well. Poverty and income inequality also induce poor mental health via multiple material and psychosocial channels [30, 31].

## Conclusions

The study concludes that the functional status of SHGs appeared to be associated with duration and type of support. SHGS that received support for year two and three were performing

better and had regular engagement with governmental organization and community members than those who received support for only year one. The performance was largely due to capacity building training that they received from BasicNeeds-Ghana. These capacity building sessions improved their leadership and negotiation skills to better engage with other stakeholders to discuss issues related to their socio-economic and health needs. In addition, SHGs that received support for more than one year are more likely to believe that active participation and engagement of SHGs is an inalienable right that should be protected to ensure fair representation in decision making. To sustain group development and to achieve growth in self-help activities, pathways for strategic support and capacity-building need to be in place at the start of the set-up. The SHGs should be linked to the local government structures to ensure regular interaction and monitoring of their activities. In addition, professional assistance, public awareness of mental illnesses, social and financial support by the government, private sector, and NGOs are important in addressing these challenges. Furthermore, the study highlights that being a person with mental health conditions in Ghana is not only about medical aspects of the condition, but more importantly about social inclusion. The medical care for people with mental health conditions are being hampered by the exclusion of psychotropic medicines in the NHIS.

## Supporting information

**S1 Data. Anonymized transcripts.**
(PDF)

## Acknowledgments

This research was conducted by BasicNeeds-Ghana (BNGh) with technical assistance from Ghana Somubi Dwumadie and the funding support of UK Aid from the British People.

## Author Contributions

**Conceptualization:** Adam Dokurugu Yahaya, Lyla Adwan-Kamara, Peter Badimak Yaro, Philip Teg-Nefaah Tabong.

**Data curation:** Philip Teg-Nefaah Tabong.

**Formal analysis:** Philip Teg-Nefaah Tabong.

**Funding acquisition:** Adam Dokurugu Yahaya, Lyla Adwan-Kamara.

**Investigation:** Lyla Adwan-Kamara, Peter Badimak Yaro.

**Methodology:** Adam Dokurugu Yahaya, Philip Teg-Nefaah Tabong.

**Software:** Philip Teg-Nefaah Tabong.

**Supervision:** Adam Dokurugu Yahaya, Philip Teg-Nefaah Tabong.

**Validation:** Philip Teg-Nefaah Tabong.

**Writing – original draft:** Philip Teg-Nefaah Tabong.

**Writing – review & editing:** Lyla Adwan-Kamara, Peter Badimak Yaro.

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
