## [Decision Letter · Decision Letter 0]

24 Sep 2024

PMEN-D-24-00352

Does supporting Self-Help Groups of people with mental conditions for longer duration lead to more effective groups? A qualitative evaluation in Ghana

PLOS Mental Health

Dear Dr. Tabong,

Thank you for submitting your manuscript to PLOS Mental Health. After careful consideration, we feel that it has merit but does not fully meet PLOS Mental Health’s publication criteria as it currently stands. Therefore, we invite you to submit a revised version of the manuscript that addresses the points raised during the review process.

We look forward to receiving your revised manuscript.

Kind regards,

Erica Breuer

Academic Editor

PLOS Mental Health

Journal Requirements:

https://journals.plos.org/mentalhealth/s/figures 

https://journals.plos.org/mentalhealth/s/figures#loc-file-requirements 

2. Figure 1: please (a) provide a direct link to the base layer of the map (i.e., the country or region border shape) and ensure this is also included in the figure legend; and (b) provide a link to the terms of use / license information for the base layer image or shapefile. We cannot publish proprietary or copyrighted maps (e.g. Google Maps, Mapquest) and the terms of use for your map base layer must be compatible with our CC-BY 4.0 license. 

Additional Editor Comments (if provided):

Thank you for submitting your manuscript.

During the revision, it may be worth considering how you have operationalised the RE-AIM in this paper (including effectivenness) and whether you have shown clear evidence for these components in the paper.

Reviewers' comments:

Reviewer's Responses to Questions

**Comments to the Author**

1. Does this manuscript meet PLOS Mental Health’s publication criteria? Is the manuscript technically sound, and do the data support the conclusions? The manuscript must describe methodologically and ethically rigorous research with conclusions that are appropriately drawn based on the data presented.

Reviewer #1: Yes

Reviewer #2: Partly

2. Has the statistical analysis been performed appropriately and rigorously?

Reviewer #1: Yes

Reviewer #2: N/A

3. Have the authors made all data underlying the findings in their manuscript fully available (please refer to the Data Availability Statement at the start of the manuscript PDF file)?

Reviewer #1: No

Reviewer #2: Yes

4. Is the manuscript presented in an intelligible fashion and written in standard English?

Reviewer #1: Yes

Reviewer #2: Yes

5. Review Comments to the Author

Reviewer #1: The article is a well written scientific article. As the authors have used the RE-AIM framework in analysing the results, I believe that the discussion of the results should be presented in the context of the RE-AIM framework. Further, there seems to be large amount of data that has been collected, however the amount of data does not get reflected in the paper and data underlying the findings has not been attached as a supplement. It would be good to have a conceptual framework at the end of the results to understand how the RE-AIM framework was used to understand the results. I suggest major revisions for this paper

Reviewer #2: Thank you for this manuscript on an important topic that presents a qualitative evaluation on whether supporting Self-Help Groups of people with mental conditions for longer duration leads to more effective groups.

Page 2-3 lines 91-97

It is unclear how the 272 groups were selected initially, more information on the identification and selection of the self-help groups will be helpful.

Page 3 line 95-96

Please add the criteria used to determine which SHGs only received additional support for the second and third years.

Page 5 lines 148-151

Please add more information on what criteria was used to determine who would continue to receive support in year two and three.

Page 5 lines 150-151

How did the researcher mitigate against bias if selection of members included those that had good knowledge on the operations of the groups?

Page 6 line 210

The author indicates that NVivo 13 was used for data analysis however the abstract indicates that NVivo 14 was used please clarify which version was used

The themes in the Results section could be organised better to make it easier for the reader to follow, there seems to be information that is being highlighted about the frequency of the meetings that participants attended, stronger relationships or partnerships with other agencies including government agencies, the rights and advocacy of participants and their inclusion in meetings however this does not come out clearly when reading the section.

Perhaps renaming the sub-headings in the section based on the most prominent theme will be helpful.

There appears to be inconsistencies in labelling the participants to indicate which groups they belong to, it will be helpful for the authors to relook at this and lable participants consistently.

Page 10

Findings on this page seem to be referring to barriers to attending meetings and it follows on from a section that explains benefits of campaigns – organising this section better to include similar themes will allow for a more logical flow.

Page 10-11

Some of the quotes used have also been used on page 9, it might be helpful to avoid repeating the quotes that have already been used, organising the themes better could help avoid this.

Pages 10-11

Sections on sustainability of SHG activities and challenges in sustaining activities seem to duplicate some concepts

Page 12 Table 2 could be moved towards the end of the article as a summary of the challenges and propose solutions for all the themes that have been highlighted

The discussion could be better organised to match the findings section to improve the flow of the section.

The reader is left wondering if a qualitative evaluation is the best way to assess effectiveness and if the study has highlighted the effectiveness of SHG through its findings, or if the study has highlighted barriers and facilitators of SHG ?

6. PLOS authors have the option to publish the peer review history of their article (what does this mean?). If published, this will include your full peer review and any attached files.

**Do you want your identity to be public for this peer review?** For information about this choice, including consent withdrawal, please see our Privacy Policy.

Reviewer #1: **Yes: **Aarti Jagannathan

Reviewer #2: No

---

## [Editor Report · Decision Letter 1]

9 Jan 2025

Does supporting Self-Help Groups of people with mental conditions for longer duration lead to more effective groups? A qualitative evaluation in Ghana

PMEN-D-24-00352R1

Dear Dr. Tabong,

We are pleased to inform you that your manuscript 'Does supporting Self-Help Groups of people with mental conditions for longer duration lead to more effective groups? A qualitative evaluation in Ghana' has been provisionally accepted for publication in PLOS Mental Health.

Best regards,

Karli Montague-Cardoso

Staff Editor

PLOS Mental Health